# Use of Technology to Promote Health and Wellbeing of People Who Are Homeless: A Systematic Review

**DOI:** 10.3390/ijerph18136845

**Published:** 2021-06-25

**Authors:** Vanessa Heaslip, Stephen Richer, Bibha Simkhada, Huseyin Dogan, Sue Green

**Affiliations:** 1Department of Nursing Science, Faculty of Health and Social Sciences, Bournemouth University, Poole BH12 5BB, UK; vheaslip@bournemouth.ac.uk (V.H.); sgreen@bournemouth.ac.uk (S.G.); 2Department of Social Work, Stavanger University, 4021 Stavanger, Norway; 3Department of Nursing and Midwifery, University of Huddersfield, Huddersfield HD1 3DH, UK; b.d.simkhada@hud.ac.uk; 4Department of Computing and Informatics, Faculty of Science and Technology, Bournemouth University, Poole BH12 5BB, UK; hdogan@bournemouth.ac.uk

**Keywords:** homeless, technology, mobile health, social exclusion, marginalised, vulnerable

## Abstract

Background: People who are homeless experience poorer health outcomes and challenges accessing healthcare contribute to the experienced health inequality. There has been an expansion in using technology to promote health and wellbeing and technology has the potential to enable people who are socially excluded, including those who are homeless, to be able to access health services. However, little research has been undertaken to explore how technology is used to promote health and wellbeing for those who are homeless. This review aims to address the questions: ‘what mobile health (mHealth) related technology is used by homeless populations’ and ‘what is the health impact of mobile technology for homeless populations’? Methods: An integrative review methodology was employed. A systematic search of electronic databases was carried out between 4 January 2021 and 30 April 2021, searching for papers published between 2015 and 2021, which yielded 2113 hits, relevant papers were selected using specified inclusion and exclusion criteria reported using the Preferred Reporting Items for Systematic reviews and Meta-Analysis. The quality assessment of each paper included in the review was undertaken using the Mixed Methods Appraisal Tool. Results: Seventeen papers were selected for review and thematic analysis identified four themes: technology ownership, barriers to use, connectivity and health benefits. Conclusion: It is evident that technology has the potential to support the health and wellbeing of individuals who are homeless; however, there are challenges regarding connectivity to the internet, as well as issues of trust in who has access to personal data and how they are used. Further research is needed to explore the use of health technology with people who are homeless to address these challenges.

## 1. Background

The term homelessness is an umbrella term for a number of groups, such as those living on the streets (rough sleepers), living in temporary accommodation and those staying temporarily with friends/family, known colloquially in the UK as “sofa surfing”. Edgar et al. [1] identified six different groups when categorising those who are homeless (Table 1) illustrating the range of categories used in defining homelessness. It is difficult to identify the prevalence of homelessness as there is currently no internationally agreed method of measuring homeless; however, 24 out of 28 European Union countries report that homelessness has increased over the last decade [2]). This pattern is also apparent in the UK, with 1768 people identified as homeless in 2010, rising to 4677 in 2018 [3].

The reasons why individuals become homeless is multifaced and can include poverty, lack of educational support, poor physical and mental health, lack of family and social support, unstable housing, lack of stable employment opportunities, trauma and abuse and incarceration [4]. Experiences that can lead to homelessness may also perpetuate it. Homelessness is inextricably linked to social exclusion as individuals are often denied the right to participate in economic, political, social and cultural life [5]). For example, reduced social rights and social integration due to living on the streets, as well as reduced political rights as lack of address and access to television leads to lack of awareness regarding political strategies and means to be able to register your political vote. This is sustained due to low social status and the cycle of poverty [6]. Lack of stable housing also contributes to social exclusion as many of the support systems designed to assist individuals require an address. Homelessness is associated with numerous United Nations Sustainable Development Goals [7] including no poverty, zero hunger, good health and wellbeing, clean water and sanitation and reduced inequalities.

### 1.1. Health and Homelessness

Interventions to promote good health include multi-agency interventions, psychosocial support and disease prevention, as well as gendered tailored interventions [8]). Public Health England outline that no single intervention is effective but rather a system-wide integrated approach is needed to meet the complex needs of individuals. However, a literature review by Clifford et al. [9] exploring homeless health and polices across four countries noted a focus on health utilisation rather than the broader public health framework which identifies structural social determinates of health of homelessness.

Health outcomes are worst for those individuals sleeping rough [10]. In the UK, the mean age of death for males who are homeless is 47 years and for females 43 years compared to a mean average age of death for people living in homes of 76 and 81 years for men and women, respectively [11]. Fazel et al. [12] noted that whilst standardised mortality ratios vary between countries, those who are homeless typically have a 2 to 5 times higher age-standardised mortality ratio than those not homeless and little has changed over the last two decades. People who are homeless can have very complex health needs, due to a combination of poor physical and mental health and addictions [13]. Furthermore, many people who are homeless struggle to access community care services and tend to access healthcare through emergency departments both in the UK [13] and internationally [14]. This lack of access to primary care services means opportunities for pro-active, preventative health care is reduced, which undoubtedly contributes to higher mortality rates.

### 1.2. Technology and Health

Whilst digital health is not a new concept, its implementation has been rather modest [15]. Despite this, digital technology was argued by the Topal Review as a new means of addressing 21st century health care challenges [16]. The recent COVID-19 pandemic resulted in many countries across the world entering some form of national lockdown [17] to reduce transmission rates. This has resulted in a shift from the traditional face to face delivery of healthcare towards a massive expansion in utilising digital technology [18] to continue to provide health care services. Whilst this expansion of digital heath care is positive for many, it has raised issues of digital inequalities for some socially excluded groups, which include both physical barriers in a lack of access to equipment, as well as educational barriers in not being able to use the technology. If these areas can be addressed, then digital access to healthcare has the potential to enhance healthcare access for socially excluded groups. Two published reviews examining the impact of information communication technology (ICTS) on the homeless and their health have been identified [19,20]. These reviews indicate that there is a lack of information regarding the use of healthcare technology by those who are homeless and the impact it has upon their health. This systematic review aims to address this by locating and evaluating the published research on the uptake of healthcare-related technology by homeless populations and the impact this technology has on health outcomes. The research questions which framed the review were (1) ‘what mobile health (mHealth) related technology is used by homeless populations’ and (2) ‘what is the health impact of mobile technology for homeless populations’?

## 2. Methods

An integrative review methodology was employed. Integrative reviews are the broadest type of research review methodology, allowing for the inclusion of experimental and non-experimental research and qualitative and quantitative studies in order to have a full understanding of the phenomenon of interest [21]. The review process was presented using the Preferred Reporting Items for Systematic Reviews and Meta-Analyses (PRISMA) 2009 Checklist. Step one of the review process (problem identification) has already been presented in the background and review questions.

### 2.1. Search Strategy

A systematic search of the literatures was conducted. Between 4 January 2021 and 30 April 2021, 10 databases were searched: Cochrane library (Cochrane systematic reviews), Academic Search Ultimate, Medline, CINAHL, SCOPUS, PsychInfo, Cochrane, Google Scholar, Eric and Web of Science. Searches were developed using keywords and database specific subject headings (Table 2). Database limiters included peer reviewed and papers written in English.

The literature search results were exported, managed and shared using EndNote bibliographic referencing software. The data evaluation phase consisted of two stages. The first stage included reviewing the record’s title and abstract [SR] against the inclusion/exclusion criteria (Table 2) and these were filed as ‘include’ or ‘exclude’. Records filed to ‘exclude’ were randomised and 10% of these were reviewed by another member of the research team [BS, HD] to confirm selection process. The second stage included a full text review of the ‘include’ results [SR] to identify the final studies for inclusion in the review. Again, records filed to ‘exclude’ were also randomised and 10% were each reviewed by another member of the research team ensuring quality assessment of the papers.

### 2.2. Data Evaluation

Characteristics of papers included in the review were summarised (Table 3). Each study was critically appraised [SR, BS] using the Mixed Methods Appraisal Tool (MMAT) [22] and 10% randomly selected were independently reviewed [VH]. Each study received an overall rating of strong, moderate or weak in accordance with MMAT, a rating of strong was given when papers met six or more of the criteria and a rating of weak was given when papers only met one or two of the criteria. 

### 2.3. Data Analysis

In stage four of the integrative review process [21], data were analysed using Braun and Clarke’s [40] process of thematic analysis in order to identify key themes. This included reading each paper in turn to familiarise ourselves with the data and to generate initial codes and categories. Once codes and categories had been identified for each paper, initial themes were identified. The initial themes were further reviewed across all of the papers included in the review and modified as necessary. These final analytical themes were shared alongside the initial codes and categories with the whole research team to ensure the credibility of the analytical process. This provided a narrative thematic analysis of the homeless population’s access to mobile health technologies and the efficacy of these technologies in improving health and wellbeing in this population.

## 3. Results

The search strategy identified 3089 records, 2113 were screened and 2072 excluded as they focused on IT solutions on housed populations or non-technological interventions. Stage two included reviewing 41 full-text articles and, at this stage, a further 24 papers were rejected, resulting in 17 papers being included in the review (Figure 1).

### 3.1. Study Characteristics

Studies included in the review included qualitive (*n* = 10), quantitative (*n* = 5) and mixed methodological designs (*n* = 2). The majority of the studies were undertaken in the United States of America (USA) (*n* = 13), followed by Canada (*n* = 2), Italy (*n* = 1) and the UK (*n* = 1).

### 3.2. Themes

The research questions framing the review were ‘what mobile health (mHealth) related technology is used by homeless populations?’ and ‘what is the health impact of mobile technology for homeless populations?’ Four main themes were identified inductively through thematic analysis: mobile phone ownership and usage, barriers to use, social connectedness and health benefits.

### 3.3. High Level of Mobile Phone Ownership and Usage

This key theme illustrates that the majority of homeless participants across all studies owned a mobile phone/smart phone. This ranged from 53% [39] to 100% [23]. The largest study (*n* = 421) identified that 94% of participants owned a mobile phone [36] and across the 17 studies included in the review (*n* = 1507), 80% (*n* = 1205) owned a mobile phone. Male and female phone ownership was very similar across all studies. However, age appeared to be a major factor in mobile phone ownership and use [34,35]. Raven et al. [34] identified that older homeless people were found to have significantly lower smartphone ownership and internet access than adults in the general population aged 65 years or more, whereas Reitzes et al. [35] identified that nearly half of homeless 45- to 54-year-old individuals did not own a mobile phone compared to 31% of 18- to 44-year-old individuals. No significant difference in mobile phone ownership was found between countries in this review but various factors including the date of publication and varying participant circumstances made comparisons uninformative.

Although mobile phone usage was high, the sophistication of the technology used varied; some participants had smartphones whilst others had those with more basic functionality, such as texts and voice-calls only [24]. Jennings et al. (2016) found that, of the 93% of participants (*n* = 41) who owned a mobile phone, 84% had internet access; the remainder used texts and calls only. Moczygemba et al. [33] identified that, although 91% (*n* = 255) of participants had text messaging available, this dropped to 59.1% for devices supporting applications. Raven et al. [34] found similar differences with 72.3% of participants with a mobile phone; only half (32.1%) of those who owned phones had smartphone functionality. Lastly, Jennings et al. [31] found a drop off between basic phone and smartphone from 93 to 84%, respectively, whilst Rhoades et al. [36] identified, overall, 94% (mobile ownership) but with only 51% owning a smartphone. Although some studies simply reported ownership of a phone without identifying phone functionality, the majority seemed to concur that a high percentage of the homeless population owned a phone and about a half to three quarters of these had smart functions. However, the way homeless participants used their phones varied [31,33,34,36]. Unsurprisingly, the main usage of mobile phones was to make calls (100%) and send texts (ranging between 72–93%) and general internet access (ranging from 22–84%). However, Von Holtz et al. [38] found that, while experiencing homelessness, participants experienced a 68% reduction in their likelihood to access the internet, compared to when they were housed. Frequency of mobile phone use also varied across the studies. Rhoades et al. [36] identified that 85% of participants used their phone daily (*n* = 261), Ritzes et al. [35] found this to be around 75% and Moczygemba et al. [33] found this to be 76%. However, in all three studies, this usage dropped dramatically when assessing daily internet use, to 39%, 17% and 24%, respectively. 

### 3.4. Barriers to Use of Mobile Phones

This theme focusses on the multitude of barriers to use of or mobile phones that were reported. Homelessness inevitably brings with it many practical implications that make the maintenance and use of mobile phones more difficult than it would be for the general population. This included battery life due to difficulties in accessing charge points [23,30,32]. In addition to charging points, Harris et al. [30] and Glover et al. [28] also identified lack of access to data as major hurdles to technology use:


*“I don’t think there’s enough services around to offer people the chance to use the internet for free … you can’t be like, ‘well, you need to do this, but we’re not going to give you any access to do that, so good luck with that,’ kind of thing” *
*(Adam, homeless, aged 18–24) [30]*

Adkins et al. [23] noted that when participants were in one place with access to a power socket for any length of time, they would take advantage of the situation and charge their phones. Their study also identified phone damage as problematic, with cracked screens and water damage being the most common problems. Four papers included in the review identified issues regarding theft of mobile phones [28,30,31,34] noted that participants were anxious about the possibility of theft whilst in shelters. Raven et al. [34] found that 47% (*n* = 277) of participants had had their phone stolen at some point, 12% on at least three occasions. Furthermore 47% had lost their phone at least once. Jennings et al. [31] identified that, in addition to theft, having a mobile phone increased experiences of harassment. As such, both theft and harassment were identified as potential sources of stress when owning a mobile phone.

Poor IT skills amongst homeless populations has been implicated in poor mental health outcomes. Harris et al. [30] found age to be key socio-demographic variation affecting homeless people’s use of technology. Participants felt that the shift in the UK to a more digital benefits system (something mirrored across many benefits systems globally) had assumed that users were well versed with information technology, although this may not be the case.


*“At the job centre there’s only one computer and that’s just for job search not to train you up on computers … it’s like you’re supposed to know everything like you were born with technology in your head”*
*(Victor, homeless, aged 55–64) [30]*

Another identified barrier to mobile phone use is that of trust. Adkins et al. [23] noted that homeless youth expressed concerns about tracking and the use of information. The participants understood the benefits and reasons for the tracking and tracing of information but were concerned about what information was being shared and why. There were significant concerns about sharing information with strangers, particularly information corresponding to phone calls and text messages. Asgary et al. [24] found that participants did not consider free-phones and data plans would be useful and cited a lack of trust in the government and system as identified below:


*“I wouldn’t take one of those, I’d rather buy myself a plan. The government never did anything for me before, why do they wanna give me a free phone?”*
(M55) [24]

Jennings et al. [31] found that some young homeless individuals viewed mobile phones as harmful. There was a feeling of distrust in the privacy levels of mobile devices, and a fear that there could be the tracing of people without permission. There was also concern that peers sharing an individual’s phone could result in being implicated in someone else’s criminal activity as a result of tracing.


*“‘It’s tapped nowadays. You can’t say nothing over the phone. They’ll track that piece in your phone! And you goin’ down’ Male, homeless, 24”*
*[31]*

Furthermore, data from 286 homeless individuals, identified 27% as having privacy concerns with the use of mobile technology and healthcare reminders [33]. Finally, Watson et al. [39] highlighted the systemic difficulties associated with trust and privacy. Certainly, the high turnover of phones and phone numbers within the homeless population makes privacy a challenging issue. This is particularly pertinent when attempting to address issues which, by their very nature, must be individually targeted and confidential information revealed, such as medication compliance or healthcare appointments.

### 3.5. Social Connectedness

Social connectedness, evident in five studies, describes the potential of the mobile phone to maintain social connections [23,26,29,31,34]. Adkins et al. [23] found that participants (homeless youth) predominantly used their phone for communication with friends and family via text and calls. In addition, the participants identified music, entertainment and social media as being useful mediums to facilitate connections with the wider world. Calvo and Carbonell [26] further examined the notion of social connectedness with regard to the homeless population using a Facebook app. They identified that, of the 33 people using social networking sites, 88% stated that their principal motivation for using these was to communicate with another person: 57% with family and 30% with friends. The results also indicated that the number of weekly hours devoted to using Facebook was a predictor of an increase in social skills and self-esteem scores. Raven et al. [34] identified that mobile technologies reduced social isolation. In this study, participants with current or prior access to a phone (*n* = 277) reported using phones to contact relatives (82.3%) and friends (77.6%). Ritzes et al. [35] identified no age group differences in using cell phones to contact family members or old friends, although homelessness in younger age groups tended to use mobile phones more to maintain ties with new friends. Jennings et al. [31] found that mobile phones were considered by the homeless population as being beneficial, primarily using them as a mechanism for communicating with others and obtaining social support:


*“I never dropped it [mobile phone], never had a cracked screen, nothin’ else … Cause I can’t live without my phone … I just can’t … I just gotta have my phone or some type of way I could communicate with somebody”*
*—Female, Age 20 [31]*

This review identifies the utility of mobile health (mHealth) technology in maintaining social contact with one’s peers. Across all age groups, homeless individuals find the facility of contacting friends and family by text and phone call a positive experience. Furthermore, studies, including those of Calvo et al. [26] and Ritzes et al. [35], have found a common usage for video streaming, music sites, social media sites and chat rooms.

### 3.6. Health Benefits

The notion of *connectedness* offers insight into the effectiveness of technology and the indirect effect on the health of homeless populations. However, many of the studies in this review (i) assessed attitudes to the use of technology as a tool to impact directly on health outcomes and (ii) examined how effective these tools can be. This concept forms the fourth theme. Curry et al. [27] examined health-advice seeking behaviours associated with the homeless and technology and identified that linked health problems, particularly mental health and addiction issues, with the frequency of those seeking advice online. The results indicated service use, and the use of the internet for stability-seeking purposes were significantly associated with race, hard drug use, and becoming homeless due to mental health problems. In fact, the use of class A drugs was associated with a twofold increase in health advice-seeking behaviour online. In addition, young homeless people who indicated they had a mental illness were five times more likely to seek help online. Conversely, the authors identified that, for every month someone is homeless, the chance of them seeking health advice online reduces by 2%.

Adkins et al. [23] found that participants reported they would be interested in using a mobile phone application for general emotional support. They also spoke of using the app to help with making day-to-day decisions, problem solving, general health advice and help with life decisions.


*“Yeah, if you’re having bad day, send a quick text, tell them the day’s been kinda rough, send you like a text like hey, you want to talk, you know?”*
*Homeless youth. [23]*

Atkins et al. [23] also noted that participants were positive about using a mobile phone to get advice and help addressing issues, such as depression, anxiety, self-harm, abuse, substance use, emotional problems, insomnia, and stress. Support was found for push notifications though an application, such as quotes, texts and videos. Further interest was found in notifications offering positive reminders, support messages and help managing relationships.


*“Some people like motivation. Some people like to feel supported. A lot of people out here don’t have a support system, so some people are looking for someone to be like hey, keep this throughout the day, try not to complain for 24 h”*
*Anon. [23]*

Three papers in this review focused on the use of technology for appointment reminders [23,24,25]. Asgary et al. [24] found that *mHealth* (mobile technology for providing health information or services) was recommended by the homeless population for reminders regarding preventive care and medical appointments to improve adherence and receive health education. The authors found that mobile technologies could potentially provide a platform for targeting health services. Appointments and prescription reminders were also found to be of use by Moczygmeba et al. [33], who examined the feasibility of using mobile technology with the homeless to communicate appointments and medication needs/changes. Again, the responses were positive to the idea of such technology; 70% felt that appointment reminders would be useful, and 60% said this for refill reminders (repeat prescriptions) and medication-taking reminders. Those who expressed most interest in reminders were found to have a history of running out of medication and missing appointments and felt this was a system that could help circumvent similar mishaps in the future. Similarly, Watson et al. [39] examined the medication behaviours of the homeless living with a mental illness and, as a secondary outcome, looked at the potential utility of technology in this. This qualitative study noted that the ability to remain connected to various supports, healthcare or otherwise, was identified as crucial by participants. However, participants had mixed options about being prompted via text message to take medication.


*“A text message to take my meds? Yes. Please. Sign me up for that”*
*(Participant 12) [39]*


*“If they text messaged me, I’d find that odd. I just didn’t find the need to do that with them. I don’t know, is it unprofessional?”*
(Participant 10) [39]

Jennings et al. [31] identified that young homeless individuals valued mHealth content relating to sexual, reproductive, and mental health. The caveat was that mobile communication was confidential, empowering and integrated with other digital media. Integrating hidden phones, financial support, and safety management were also identified as avenues to improve homeless youths’ access to and engagement with mHealth strategies over time. In an older population, McInnes et al. [32] found that participants (homeless veterans) felt that mobile-phone calls or text messages could be used to remind patients of appointments, prescription refills, medication taking, and returning for laboratory results. The use of mobile phone text messages aided the participants in staying organised and made the storage of information easy and convenient (times of appointments were saved in messages). Cost remained an issue, as was concerns about receiving excessive texts, but the consensus was positive for the use of mobile phones for health-related purposes.

Glover et al. [28] assessed the acceptability of delivering automated mental health resources via smartphone technology. The technology was specifically designed for the study, the *Pocket helper 2.0* (Center for behavioral therapies, Chicago, IL, USA). Results were largely positive, as 63% of respondents reported benefiting from the intervention. Daily tips were very useful by participants, as were the signposting to up-to-date resources and the automated self-help system. Interactive features, such as the telephone hotline and emotional support tool, proved less popular. It seemed that participants appeared to value the more practical and direct application features.

Attitudes towards technology utility as a health-advice tool were positive across the review; the homeless populations, with a few reservations, reported that technology could play a part in their lives. However, research directly assessing the health impact of technology in the homeless population is less prevalent. One such study, investigated the feasibility of a remotely delivered mental health intervention for homeless young adults (18–24) through apps designed to improve mental health through behavioural change [37]. Researchers measured participants levels of depression, emotional regulation, and PTSD throughout the trial. The participants’ recorded high levels of satisfaction with the apps during the trial. Although there were no clinically significant changes (the study was not designed or powered to achieve this) in depression, emotional regulation or PTSD, small improvements in all three were noted with small effect sizes. However, the intervention was small both in time frame (1 month) and participant numbers (*n* = 35). The utility of this kind of health intervention was, nonetheless, demonstrated to some extent by both positive engagement and positive outcomes, but further research is needed.

## 4. Discussion

Homelessness is a worldwide public health concern due to the associated health inequalities [42]. This review sought to address two main questions ‘what mobile health (mHealth) related technology is used by homeless populations?’ and ‘what is the health impact of mobile technology for homeless populations?’ It notes the potential of utilising digital technology to promote health and wellbeing; however, for this to occur, we need to address the problems associated with accessibility and usability. At first glance, the results of this review are encouraging. Certainly, a large percentage of the homeless population owns a mobile phone, and the majority of those have smart functionality. However, the practical implications of homelessness quickly take a toll on using this technology effectively. Firstly, mobile phones are expensive and sellable, making them an easy target for theft. As well as this, upkeep is not cheap, data plans are not always affordable and Wi-Fi not consistently available. Additionally, charging points are few and far between, so battery life is a constant problem for many, as well as damage due to damp conditions. Due to this, people are who are homeless are recognised as digitally excluded (NHS undated), which in turn hampers their ability to access services designed to help them find more stable accommodation [43] perpetuating their homelessness. Yet little, it appears, is being done to address this, as the majority assume (incorrectly) that everyone has access to the internet [21]. The Office for National Statistics [44] identified that, in 2018, there were 5.3 million adults (10% of the adult population) in the UK who have either never used the internet or not accessed it in the last three months; we argue that many of these individuals will be homeless. An additional issue in utilising technology to address health inequalities experienced by the homeless is competency, particularly organising benefits and accessing financial assistance online. Older people who are homeless felt further marginalised by the modern benefits system that ‘assumes’ digital competence and confidence. Making sure that individuals are trained correctly is vital, and Greer et al. [45] argue this must be through a personalized learning format.

Trust, an issue pertinent in homeless populations, is an important area to consider in using digital technology to promote health. This review identified frequent references to a lack of trust, including fears of tracking and tracing one’s whereabouts, abuse of data sharing between by mobile devices, and a general mistrust of the healthcare systems with regard to information. This lack of trust reflects a wider lack of trust in people who are homeless using healthcare services. Van den Berk-Clark and McGuire [46] argue that the issue of trust between people who are homeless and clinical staff is multi-faceted and is influenced by technical competence as well as the degree the individual is made to feel welcome by the service. As such, any growth in online health provision without addressing the digital exclusion of those who are homeless is likely to further damage any trust and, therefore, perpetuate existing poor healthcare access.

The review identified that mobile phone use is extensive across the homeless populations studied. There is a feeling across the papers reviewed that this is unexpected or surprising. It perhaps should be neither. A need to be contactable and to communicate is not exclusive to those with homes and it may serve future research well to approach these needs as ubiquitous rather than unique to any particular population. The notion of connectivity through social media and text messaging was identified as very important for the homeless population, and we argue that this connectivity is potentially more important to the homeless population than the population at large, providing continued connection with friends and family that can be lost when living on the streets. The relationship between social connections and health is well established, in that our social relationships affect our mental and physical health and mortality [47]. Research by Groton and Radey [48] with homeless women, identified that social support was integral to people who were homeless, enabling them to cope with their situation whilst, conversely, a loss of support contributed to or prolonged their homelessness. As such, perhaps having a mobile phone and internet access should be seen as an essential rather than a luxury item for those who are homeless.

Considering the rise in the use of technology to improve health and the stark health inequalities experienced by those who are homeless. This review identified that studies showing the direct effect of digital technology use on health were limited. Some evidence was offered with regard to depression, anxiety, stress and PTSD [26,27,37], but trials are small and clinically significant outcomes have not been demonstrated. However, any change that did occur, albeit with small effect sizes, was in the desired direction. Furthermore, the review found a limited number of studies measuring potential health impact. One main area of technology utility measured was that of reminders for repeat prescriptions or healthcare appointments which were met with a positive response and increase in medication compliance. There is clearly much potential here, however, the degree to which this occurs is not known and worth exploring further. Nevertheless, this does not come without challenges, due to the high turnover of phones (from sale and theft) and, consequently, frequent movement of phone numbers and SIM cards, which could inadvertently lead to breaches in confidentiality and information governance issues. In addition, issues of access in a physical (internet access) and capacitive (skills to navigate) sense could also inhibit the capacity of such initiatives, especially for older people who this review noted are less likely to own a phone or be able to use the full functionality, even though they are a more likely to have more complicated healthcare needs. The UK Government has proposed GBP 1.1 million to NHS projects to support those who are digitally excluded [49]; however, this only addresses the capacity and not issues of access.

Literature around the health benefits of technology for the homeless population remains sparse and varied. On the one hand, many studies have assessed the attitudes of homeless populations to the use of digital technology for health purposes. This has mainly produced positive results. Homeless populations appear to view technology as having potential health benefits, appointment reminding and online support being consistently appealing. What seems to be lacking are any substantial attempts to explicitly demonstrate health benefits from a clinically significant perspective. There are clear and obvious difficulties in carrying out RCTs with a homeless population and transient lifestyles make longitudinal studies unpredictable with potentially high attrition rates. However, while it is encouraging that homeless populations have a positive view of digital technology and health interventions, this needs to be supported by empirical evidence of actual health benefits. 

## 5. Limitations

Studies tended to be conducted in global north countries with a predominance in America and Canada with little examination of the use of technology with the homeless in other countries and further research is needed in the UK and Europe and in lower income countries. This review also only included published academic papers written in English and, as such, we recognise there may be other research not captured in this review.

## 6. Conclusions

This review asked two questions. The first question, pertaining to use of technology amongst homeless populations, was answered with some clarity. They are owned by most, if not always utilised effectively. Difficulties remain with barriers, and these really need addressing at a system level to maximise the opportunity of using digital technology to address health inequalities experienced by people who are homeless. The second question, the heath impact of using technology, was answered only partially. Social connectedness and signposting seem to be of great value to homeless populations, but the RCT studies and empirical data are lacking. Future research would do well to focus on the gaps in the literature and the lack of data relating to health outcomes.

## Figures and Tables

**Figure 1 ijerph-18-06845-f001:**
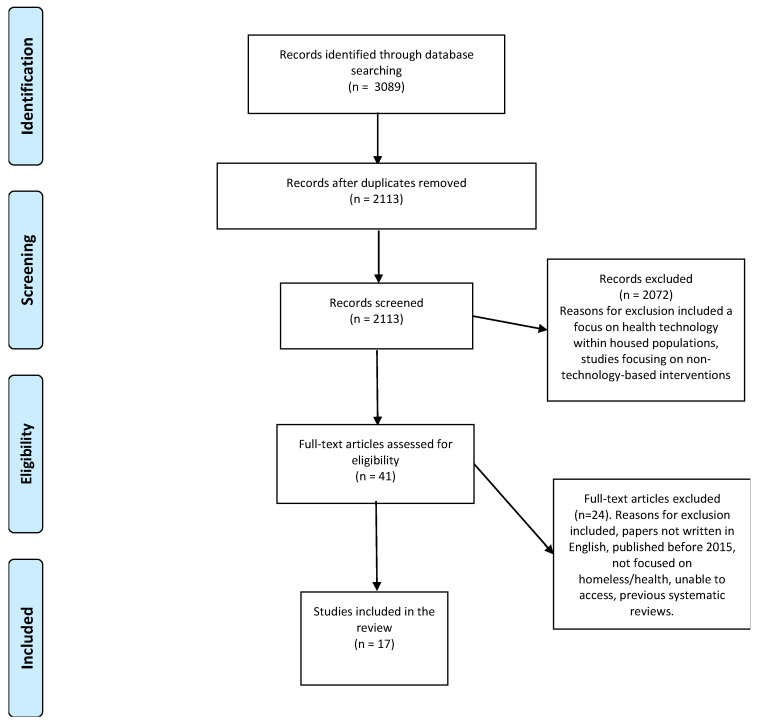
PRISMA Flow Chart [41].

**Table 1 ijerph-18-06845-t001:** Definitions and measuring homelessness [1].

	Situation	Category of Homelessness
1	People Living Rough	Primary Homeless
2	People in emergency accommodation	Secondary homeless
3	People living in accommodation for the homeless	
4	People living in institutions (due to be released but no home to go to)	
5	People living in non-conventional dwellings due to lack if housing	
6	Homeless people living temporarily in conventional housing with family and friends (due to lack of housing)	

**Table 2 ijerph-18-06845-t002:** Search terms used/inclusion and exclusion criteria.

**Search Terms** Population: Homeless OR rough sleeper OR sofa surfer OR emergency supported housing OR without shelter OR roofless OR people without shelter OR street people OR without homes OR people on the street OR night shelter OR squatters Exposure: Cell phone OR Mobile phone OR Tech* OR app* OR ICT OR Digital OR software OR computing tools OR online OR Persuasive Tech* OR Assistive Technology OR Web based OR Web based OR e-health Outcome: Health OR Wellbeing OR Behaviour Changes OR Healthcare services OR illness OR follow up OR welfare OR access to services	
**Inclusion** Peer Reviewed English Language Considers homeless population and mHealth (mobile computing, medical sensor, and communication technologies for healthcare) Papers published from 2015 to 2021 Empirical studies Participants 18 years old or more	**Exclusion** Editorials/Review/Commentary papers Conference abstracts Technology not focused on homeless population Gypsy/Roma Travellers living on the roadside Children Grey literature (Literature published outside of library holdings or institutional repositories)

**Table 3 ijerph-18-06845-t003:** Studies included in the review.

Author, Year, Country	Aim	Methodology (Sample, Methods, Analysis)	Findings	Quality Appraisal Score
Adkins et al. [23] 2017 USA	To understand homeless youths’ use of technology, mental health experiences and needs.	Grounded Theory study. In total, 5 focus groups with homeless youths (*n* = 24). Thematic analysis.	Reported using mobile phones frequently for communication, music, and social media. However, practical problems with battery life, charging, theft and breakages. Lack of trust and history of poor relationships with mental health providers. Mixed feelings regarding technology which shared information with a provider. Researchers recommend the development of mental health interventions focusing on technology-based treatment options.	MMAT—strong
Asgary et al. [24] 2015 USA	To explore perceptions, attitudes, and experiences of the homeless with regard to potential mHealth *(mobile technology for providing health information or services*) methods. To assist in design programs to mitigate some of these barriers and address health disparities among homeless individuals.	Qualitative study. Semi-structured individual interviews (*n* = 50). Qualitative descriptive approach.	78% owned a mobile phone and text messaging improved healthcare through appointments reminders. A positive attitude toward mobile technologies could be effectively used to improve health education, preventive care and chronic disease management. Policies to improve access to mobile technologies along with targeted mHealth strategies considered for mobile homeless populations.	MMAT—strong
Buccieri & Molleson [25] 2015 Canada	To design and develop a mobile application for homeless youth.	Participatory qualitative study (*n* = 12).	Developing the application allowed participants to express their opinions and frustrations and provided opportunity to make the lives of their peers better through improved access to supports and services.	MMAT—moderate
Calvo and Carbonell [26] 2018 Spain	Comparison of a Facebook training course and an office software course and their effect on psychological wellbeing in a group of individuals experiencing homelessness.	Experimental and longitudinal study design (*n* = 92). Mixed analysis of variance of repeated measures performed.	Number of hours devoted weekly to using Facebook was a predictor of increase in social skills (B = 3.43, r^2^ = 0.405) and self-esteem (B = 0.382). Facebook could be a useful tool to improve psychological wellbeing and socialization in the homeless.	MMAT—moderate.
Curry [27] 2016 USA	To analyse substance use and self-reported reasons for homelessness. To understand the relationship between youth characteristics and behaviours and use of the Internet for stability-seeking purposes.	Nonprobability sample of homeless youth. Self-administered questionnaire during 2 periods of data collection (*n* = 642) at drop-in agencies. Bivariate and multivariate logistic regressions conducted.	Use of the internet for stability-seeking purposes are significantly associated with race, hard drug use and mental health problems. Individuals who use ‘hard’ drugs are 2 times more likely (*OR* = 1.82, *p* < 0.05) to use health services than those who do not use hard drugs. Youth who indicate they are homeless because of mental illness are 5 times more likely (*OR* = 5.13, *p* < 0.001) to look online for health services than homeless youth who do not give mental illness as a reason for homelessness.	MMAT—moderate
Glover [28] 2016 USA	To establish the feasibility (as measured by phone retention rates) and acceptability (i.e., participant ratings of resources) of delivering automated mental health resources via smartphone technology.	Qualitative study with homeless youth (*n* = 100). Self-reporting on efficacy of app. In total, 48% of participants responded to the 3-month surveys dropping to 19% at the end point survey (6 months).	23% experienced problems with the phones (e.g., theft, loss, and technological issues). 63% (30/48) to 68% (13/19) of respondents at both time points reported benefiting from the intervention. Participants reported receiving the most benefit from the daily tips and surveys. Most used features were the app providing up-to-date resources and the automated self-help system. Interactive features were the least used features and were rated as the least beneficial.	MMAT—moderate
Gui et al. [29] 2016 USA	To ascertain the current use of mobile phone technology by a homeless population.	Qualitative, ethnographic study. In-depth, semi-structured, face-to -face interviews (*n* = 14). Thematic analysis.	Participants used mobile phones and computers for managing friendships, enlisting family support, finding housing, and seeking employment. Urban communities should adopt a multi-agency approach and provide support centres offering homeless people access to computers and Wi-Fi.	MMAT—weak
Harris [30] 2020 UK	To discuss the notion that ‘the current shift to digitization fails to recognize the variation and complexity surrounding homeless people’s use of technology, with homeless people as technology users often placed into homogenizing categories.	Qualitative interviews and observations carried out with both the homeless and voluntary sector organisations. Narrative interviews (*n* = 16) were carried out across 3 homeless organisations. in England. A total of 16 narrative interviews took place with homeless people who were accessing services at the three participating organizations, and 16 semi-structured interviews with staff and volunteers.	Discrepancies in ownership and access to devices were found to impact homeless people’s ability to successfully manage benefit claims. Age affected homeless people’s use of technology to access to advice and welfare benefits. Findings suggest that when people are seeking housing or homelessness advice there is a lack of available information, particularly when first becoming homeless. Lack of trust. Most homeless participants were initially somewhat sceptical of the role of technology within the lives of homeless people.	MMAT—strong
Jennings et al. [31] 2016 USA	To assess the potential of mHealth technologies for homeless young people. To collect data on phone behaviours, perceptions, and intervention preferences among youth experiencing homelessness	Mixed-methods research design, homeless participants (*n* = 52) attended one of 9 qualitative focus group discussions and quantitative structured survey. Thematic analysis.	78% of participants had a mobile phone, and 80% had Internet, texting, and multimedia features. Common reasons for switching phones/numbers were harassment (20%), missed payments (17%), device upgrades (17%), and interpersonal conflict (12%). Mobile phones were seen as beneficial enabling communication and social support from others. Conversely, tracing was identified as a potential risk by some participants. Some participants reported a mobile phone was simply not a priority.	MMAT—strong
McInnes et al. [32] 2015 USA	Examination of homeless veterans’ access to and use of IT, attitudes toward health-related IT use, and barriers to IT in the context of homelessness.	Qualitative face to face interviews (*n* = 30). Inductive thematic analysis was used.	90% of participants had a mobile phone and were receptive to IT use for health-related communications. Common difficulty was the lack of a stable mailing address. Participants felt mobile-phone calls or text messages could be used to remind patients of appointments, prescription refills, medication taking, and returning for laboratory results. Concerns expressed re. costs and privacy.	MMAT—strong
Moczygemba et al. [33] 2018 USA	To describe homeless persons’ access and use of cell phones and their perceptions about using cell phone alerts to help manage medications and attend health care appointments.	A qualitative cross-sectional survey (*n* = 290). Logistic regression was used to examine predictors of interest in using a cell phone to manage medications and appointments.	89%, 258 participants had a cell phone. 77% percent were interested in appointment reminders. 66% in refill reminders. 60% in medication taking reminders. 54% in medication information messages. Mobile technology is a feasible method for communicating medication and appointment information to homeless.	MMAT—strong
Raven [34] 2018 USA	Aimed to describe the access to and use of mobile phones, computers, and internet among homeless adults.	A qualitative, semi-structured study with older adults (*n* = 300), aged 50 years plus.	Older homeless adults could benefit from portable internet and phone access. Barriers to mobile phone and internet use, included financial barriers, functional and cognitive impairments. 72.3% participants currently owned or had access to a mobile phone. Of those, most had feature phones, rather than smartphones (89, 32.1%), and did not hold annual contracts (261, 94.2%). Over halfhad ever accessed the internet. Participants used phones and internet to communicate with medical personnel (179, 64.6%), search for housing and employment (85, 30.7%), and to contact their families (228, 82.3%).	MMAT—strong
Reitzes et al. [35] 2018 USA	To investigate how homeless people in Atlanta access and use mobile phones, internet, and email.	A quantitative questionnaire study (*n* = 206). Statistical analysis occurred.	62% owned a cell phone in the last year, 63% used the internet in the last month or more frequently, 77% had email accounts. The number of times someone was homelessness was related to cell phone ownership and frequency of use. Age was related to computer knowledge, internet frequency, and having an email account.	MMAT—strong
Rhoades [36] 2017 USA	To assess homeless populations access to and use of mobile technology with a view to ascertain potential for healthcare interventions.	Quantitative study (*n* = 421). Descriptive findings and age-matched comparisons with general population data.	94% owned a cell phone currently. Turnover in both phone ownership and phone numbers was high. 58% currently owned smartphones; 86% of smartphones used Android operating systems. Daily cell phone use was reported by 85%, and 76% reported text messaging in the past 3 months. Daily Internet use was reported by 39%, while 33% reported no past 3-month Internet access. Slightly higher rates of current cell phone ownership than same-age persons in the general population, at 95%, compared to 90%.	MMAT—
Schueller et al. [37] 2019, USA	To develop and determine the feasibility and acceptability of engaging young adults (18–24 years) experiencing homelessness in a remotely delivered mental health intervention.	Quantitative, single-arm feasibility pilot trial (*n* = 35).	Feasible to engage homeless young adults in mental health services using technology-based intervention with high rates of satisfaction. In total, 52% (12/23) reported that they were very or extremely satisfied with their participation. Little change from pre- to post treatment on measures of depression (d = 0.27), post-traumatic stress disorder (d = 0.17), and emotion regulation (d = 0.10).	MMAT—moderate
Von Holtz [38] 2018 USA	To investigate how homeless adolescents, use mobile technologies for general and health-related purposes, whether the scope of their use changes with housing status, and their interest in a website dedicated to youth experiencing homelessness.	Mixed methods study. Qualitative semi structured interviews (*n* = 87) and questionnaire. Thematic and SPSS statistical analysis occurred.	56% accessed the internet at least once a day, with 86% accessing once a week. While experiencing homelessness, subjects reported a 68% decreased odds in internet access frequency. Semi-structure interviews (N = 10) themes (1) there were changes in internet behaviours while experiencing homelessness, (2) health status as a major concern and reason for Internet use, and (3) while experiencing homelessness, participants indicated their behaviours were more goal-oriented and less focused on leisure or entertainment activities.	MMAT—strong
Watson et al. [39] 2018 Canada	To determine medication-taking behaviours and factors influencing adherence in patients with mental illness and recent homelessness. To explore patients’ perceptions on mobile technology use to support adherence.	A qualitative description and constructivist approach study (*n* = 15).	Eight participants (53%) had access to a mobile phone. There was a moderate interest in the use of mobile technology to support medication adherence, with cost and technology literacy identified as barriers. Themes arising from the data included patient factors (i.e., insight, attitudes towards medications, coping strategies) and external factors (i.e., therapeutic alliance, family support that impacted adherence) and technology use and health.	MMAT—moderate

## Data Availability

Not applicable. No new data were created or analyzed in this study. Data sharing is not applicable to this article.

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
