# Peer review of "Use of Technology to Promote Health and Wellbeing of People Who Are Homeless: A Systematic Review"

_ijerph, 2021, doi:10.3390/ijerph18136845_

Round 1
Reviewer 1 Report
Overall – this is a very nice paper that has been well-written and is interesting. I only have a few suggestions for the authors.
Background
- The authors state, “Homelessness is inextricably linked to social exclusion as individuals are often denied the right to participate in economic, political, social and cultural life.” A few brief examples might help to illustrate this point.
Methods
- Could the authors briefly explain “integrative review methodology.”
Table 2 – need to define for table even if defined in text
- Inclusion criteria includes “Considers homeless population and mHealth.” What is “mHealth?”
- Exclusion criteria includes “Grey literature.” What is “grey literature?”
Author Response
Overall – this is a very nice paper that has been well-written and is interesting. I only have a few suggestions for the authors.
Background
- The authors state, “Homelessness is inextricably linked to social exclusion as individuals are often denied the right to participate in economic, political, social and cultural life.” A few brief examples might help to illustrate this point.
Response: We wish to thank the reviewer for their comments.
Thank you for this comment we have added “For example, reduced social rights and social integration due to living on the streets as well as reduced political rights as lack of address and access to television leads to lack of awareness regarding political strategies and means to be able to register your political vote”
Methods
- Could the authors briefly explain “integrative review methodology.”
Response: Integrative review methodology has been more fully defined.
Table 2 – need to define for table even if defined in text
- Inclusion criteria includes “Considers homeless population and mHealth.” What is “mHealth?”
- Exclusion criteria includes “Grey literature.” What is “grey literature?”
Response: In table two we have deifne the key terms (Grey literature and mHealth) as Grey literature (Literature published outside of library holdings or institutional repositories) and mHealth as ……
Reviewer 2 Report
Thanks for sending this systematic review manuscript to me. This paper focused on the use of technology to promote the health and wellbeing of people who are homeless, which is interesting and meaningful work.
This research was completed under the framework of PRISMA, which complies with the basic requirements and specifications of the review.
However, the following concerns should be well-addressed before accepting for publication :
The author should explain why the period of this study is limited to 2021? Since only papers published between 4/1/21 and 30/4/21 have been included for analysis.
The content in Table 3 seems to be too much and should be simplified appropriately.
- A small number of grammatical errors should be checked again throughout the manuscript.
Author Response
Thanks for sending this systematic review manuscript to me. This paper focused on the use of technology to promote the health and wellbeing of people who are homeless, which is interesting and meaningful work.
This research was completed under the framework of PRISMA, which complies with the basic requirements and specifications of the review.
Response: Thank you for your comments regarding how meaningful this work is.
However, the following concerns should be well-addressed before accepting for publication :
- The author should explain why the period of this study is limited to 2021? Since only papers published between 4/1/21 and 30/4/21 have been included for analysis.
Response: We have clarified in the abstract that the search of the databases occurred between 4/1/21 and 30/4/21 and added it “searching for papers published between 2015-2021”
- The content in Table 3 seems to be too much and should be simplified appropriately.
Response: Table three has been simplified
- A small number of grammatical errors should be checked again throughout the manuscript.
Response: The paper has been checked and grammatical errors amended
Reviewer 3 Report
The authors aim to discuss two questions 'what mHealth related technology is used by homeless populations’ and ‘what is the health impact of mobile technology for homeless populations’. There is a lack of explanation of detailed process of thematic analysis used in the study. Overall, moderate innovation and moderate significance.
Author Response
The authors aim to discuss two questions 'what mHealth related technology is used by homeless populations’ and ‘what is the health impact of mobile technology for homeless populations’. There is a lack of explanation of detailed process of thematic analysis used in the study. Overall, moderate innovation and moderate significance.
Response: Thank you for this comment re the lack of detailed process of thematic analysis, we have added to this to make more transparent the process linked to Braun and Clarke’s process